# Clinical Evaluation of VITEK MS PRIME with PICKME Pen for Bacteria and Yeasts, and RUO Database for Filamentous Fungi

**DOI:** 10.3390/microorganisms12050964

**Published:** 2024-05-10

**Authors:** Hyeyoung Lee, Jehyun Koo, Junsang Oh, Sung-Il Cho, Hyunjoo Lee, Hyun Ji Lee, Gi-Ho Sung, Jayoung Kim

**Affiliations:** 1Department of Laboratory Medicine, International St. Mary’s Hospital, College of Medicine, Catholic Kwandong University, Incheon 22711, Republic of Korea; shomermaid@catholic.ac.kr (H.L.); andrea5169@ish.ac.kr (J.K.); sungil4007@ish.ac.kr (S.-I.C.); ljh1107@ish.ac.kr (H.L.); dlguswl521@naver.com (H.J.L.); 2Biomedical Institute of Mycological Resource, International St. Mary’s Hospital, College of Medicine, Catholic Kwandong University, Incheon 22711, Republic of Korea; lordjs05@gmail.com; 3Department of Convergence Science, College of Medicine, Catholic Kwandong University, Gangneung 25601, Republic of Korea

**Keywords:** MALDI-TOF mass spectrometry, VITEK MS PRIME, PICKME, filamentous fungi, yeast, bacteria

## Abstract

The VITEK MS PRIME (bioMérieux, Marcy-l’Étoile, France), a newly developed matrix-assisted laser desorption ionization time-of-flight mass spectrometry (MALDI-TOF MS) system, alongside the VITEK PICKME pen (PICKME), offers easy sample preparation for bacteria and yeasts. The VITEK MS PRIME also offers two software platforms for filamentous fungi: the IVD database and the RUO database. Our study evaluated its identification agreement on 320 clinical isolates of bacteria and yeasts, comparing PICKME and traditional wooden toothpick sampling techniques against MicroIDSys Elite (ASTA) results. Additionally, we assessed the IVD (v3.2) and SARAMIS (v4.16) RUO databases on 289 filamentous fungi against molecular sequencing. The concordance rates for species-level identification of bacteria and yeasts were about 89.4% (286/320) between the PICKME and wooden toothpick, and about 83.4–85.3% between the VITEK MS PRIME and ASTA MicroIDSys Elite. Retesting with PICKME improved concordance to 91.9%. For filamentous fungi, species-level identification reached 71.3% with the IVD database and 85.8% with RUO, which significantly enhanced basidiomycetes’ identification from 35.3% to 100%. Some strains in the IVD database, like *Aspergillus versicolor*, *Exophiala xenobiotica*, and *Nannizzia gypsea*, failed to be identified. The VITEK MS PRIME with PICKME offers reliable and efficient microorganism identification. For filamentous fungi, combined use of the RUO database can be beneficial, especially for basidiomycetes.

## 1. Introduction

Identifying microorganisms typically demands significant effort, time, and resources using various methods, such as physiological, serological, biochemical, and chemotaxonomic approaches. While genomic methods offer high reliability, they are slow and require extensive expertise [1]. In contrast, matrix-assisted laser desorption ionization time-of-flight mass spectrometry (MALDI-TOF MS) offers a swift, cost-efficient alternative capable of processing large volumes of samples simultaneously [2]. This technique has revolutionized conventional identification methods in laboratories by enabling rapid pathogen identification [3]. Although it is widely accepted for identifying bacteria and yeasts, there are still challenges when it comes to filamentous fungi [4]. The main challenge is the lack of a sufficient database and of an effective and rapid protein extraction method [5]. 

Currently, two manufacturers have received FDA approval and IVD certification for their MALDI-TOF MS systems: bioMérieux (Marcy-l’Étoile, France) and Bruker Daltonics (Bremen, Germany). And two new companies have joined the market, ASTA (Suwon, Republic of Korea) and Zybio (Chongqing, China) [6]. Appropriate sample pretreatment methods are crucial for the successful identification of microorganisms in MALDI-TOF systems. Direct colony spotting and on-target extraction methods are commonly used to identify bacteria, including Gram-positive, Gram-negative, and mucinous bacteria. The in-tube extraction method, a more complex and time-consuming approach, is used for microorganisms that are difficult to identify [7]. 

The VITEK MS PRIME (bioMérieux, Marcy l’Etoile, France) is a new version of the VITEK MALDI-TOF MS instrument developed by bioMérieux. This upgraded instrument features a continuous load-and-go sample loading system, prioritization for critical patient samples on urgent slides, and new internal components, all of which are designed to enhance sample processing speed and reduce handling time [8]. The VITEK MS PRIME system utilizes VITEK MS IVD Database 3.2., but it also allows for additional analysis using SARAMIS^®^ Knowledge Base v4.16 (for research use only).

The VITEK PICKME pen (hereafter referred to as PICKME, bioMérieux) is a device with a disposable nib that facilitates the user’s ability to pick and smear colonies on the VITEK MS target slide, improving smear quality. Compared to conventional wooden toothpicks, PICKME provides better uniformity and thinner deposits [9]. A study highlighted by the American Society for Microbiology (ASM) Microbe 2019 showed its ability to decrease variability among users, reduce sample preparation time by 26–42%, and improve uniformity across different user experience levels [10].

In this study, bacterial and fungal isolates detected in clinical specimens were identified using the VITEK MS PRIME to confirm their identification capabilities. Both bacteria and yeasts were tested using the PICKME and a conventional wooden toothpick. The results were also compared with the MicroIDSys Elite (ASTA, ASTA Corp., Suwon, Republic of Korea) MALDI-TOF MS currently used in our laboratory. For filamentous fungi, clinical isolates previously identified by molecular sequencing were analyzed using VITEK MS IVD Database 3.2 and SARAMIS^®^ Knowledge Base v4.16 (RUO database), respectively, to find the clinical utility of the RUO database.

## 2. Materials and Methods

### 2.1. Clinical Isolates

A total of 320 bacteria and yeast isolates were obtained from International St. Mary’s Hospital, consisting of 268 bacteria (124 Gram-negative bacilli, 91 Gram-positive cocci, 22 Gram-positive bacilli, 16 anaerobes, 15 Gram-negative cocci) and 52 yeasts (37 *Candida* species, 15 others). Among them, 228 isolates (76 isolates from 24 species for Gram-negative bacilli, 60 isolates from 31 species for Gram-positive cocci, 14 isolates from 8 species for anaerobes, 14 isolates from 8 species for Gram-positive bacilli, 14 isolates from 7 species for Gram-negative cocci, 50 isolates from 20 species for yeasts) were selected from the list of strains recommended by Clinical and Laboratory Standards Institute (CLSI) guideline M58-ED1 [11]. The remaining 92 strains were obtained from clinical samples requested in the laboratory. All isolates were confirmed by ASTA MicroIDSys Elite (ASTA), VITEK2 (bioMérieux), and/or molecular sequencing by CLSI guideline MM18 [12].

For the evaluation of filamentous fungi, 289 fungal isolates deposited in the Korean nationwide fungal collection network by the National Culture Collection for Pathogens were used. Fungal isolates compromising 31 genera and 79 species were obtained from various clinical specimens from 10 hospitals in the Republic of Korea (Seoul, Incheon, Suwon, Uijeongbu, Busan, Daejeon, Daegu, Chuncheon, and Jeju) between March 2017 and December 2020. They were recovered from various clinical specimens, such as respiratory specimens, blood, urine, CSF, body fluids, tissue, eye, nose, ear, nail, skin, and hair. Preservation was accomplished using liquid nitrogen in a deep freezer to minimize loss. The identification of the isolates involved macroscopic phenotypic identification/microscopic examination and molecular sequencing, as described in a previous study [2].

Approval for this study was obtained from International St. Mary’s Hospital, Catholic Kwandong University College of Medicine in Korea (IS22ESSE0006).

### 2.2. MALDI-TOF MS Analysis

For bacterial and yeast isolates, the VITEK MS PRIME [13] and ASTA MicroIDSys Elite [2] systems were used according to the manufacturer instructions. Sample preparation was performed by the direct smearing method, as per the manufacturer’s instructions. The VITEK MS PRIME analysis was conducted using both the PICKME and a wooden toothpick. The spectra generated by the VITEK MS PRIME were analyzed by the VITEK MS Software (version 1.1.0—203571250) and VITEK MS IVD Database 3.2. ASTA MicroIDSys Elite experiments were conducted using the wooden toothpick application. The MS spectra obtained by ASTA MicroIDSys Elite were analyzed with the reference library (CoreDB version 1.27.04). Duplicate spreading on a single-use target slide was performed for all samples.

Identification of filamentous fungi was performed using the VITEK MS Mold Kit, following the manufacturer’s instructions. After experiments, filamentous fungi were analyzed in the IVD database first in the VITEK MS PRIME. Strains that were not identified at the species level in the IVD database were analyzed once more with VITEK MS SARAMIS v4.16 RUO database.

### 2.3. Interpretation and Analysis of the Results

For bacterial and yeast isolates, concordance was determined when both the VITEK MS PRIME and ASTA MicroIDSys Elite provided the same species or genus identification at the species or genus level. In cases of no identification or discrepant results, 16S rRNA gene sequencing was used as a reference method. For filamentous fungi, consistency was determined if one or more of the duplicate results coincided with the sequencing result. Identification failure was recorded when two results were inconsistent or identification was not possible. Statistical analysis was performed using MedCalc Statistical Software version 19.2.1 (MedCalc Software Ltd., Ostend, Belgium), using Chi-square and Fisher’s exact tests with a two-tailed *p*-value.

## 3. Results

### 3.1. Bacterial and Yeast Identification Using PICKME and Wooden Toothpicks

Using the wooden toothpicks, the VITEK MS PRIME identified bacteria and yeast at the species level in 91.9% (294/320) of cases, while PICKME achieved species-level identification in 90.3% (289/320) (Table 1). The concordance rate of wooden toothpicks and PICKME in the VITEK MS PRIME was 89.4% (286/320).

Among the 34 bacterial isolates with discrepancies between the wooden toothpick and PICKME (Table 2), 8 isolates (2.5%) were correctly identified only by the wooden toothpick, 3 (0.9%) were correctly identified only by PICKME, and 23 isolates (5.9%) showed no identification (*n* = 19) or incomplete identification (*n* = 4).

After retesting, eight isolates showed species-level agreement between the wooden toothpick and PICKME. Ultimately, 294 isolates (91.9%) showed species-level agreement between the wooden toothpick and PICKME. Additionally, two isolates showed genus-level agreement between PICKME and the wooden toothpick.

There was a total of 15 strains of *Corynebacterium* included (5 *Corynebacterium striatum*, 3 *Corynebacterium diphtheriae*, 2 *Corynebacterium amycolatum*, 2 *Corynebacterium jeikeium*, 2 *Corynebacterium afermentans*, and 1 *Corynebacterium simulans*). Among these strains, 11 were successfully identified at the species level using both the PICKME and wooden toothpick methods. However, for the two strains of *Corynebacterium jeikeium*, only the wooden toothpick method achieved species-level identification, while the PICKME method failed. Additionally, for the two strains of *Corynebacterium afermentans*, both the PICKME and wooden toothpick methods resulted in failed identification.

### 3.2. Bacterial and Yeast Identification Using VITEK MS PRIME and ASTA MicroIDSys Elite

ASTA MicroIDSys Elite achieved a species-level identification rate of 88.1% (282/320). The concordance rates for species-level identification were 85.3% (273/320) between the wooden toothpick and ASTA MicroIDSys Elite, and 83.4% (267/320) between PICKME and ASTA MicroIDSys Elite. Two isolates (*Bacteroides thetaiotaomicron*, *Stenotrophomonas maltophilia*) initially showed no identification in ASTA MicroIDSys Elite but were correctly identified upon retesting. As both isolates showed identification failure in the VITEK MS PRIME, the agreement rate between the ASTA MicroIDSys Elite and VITEK MS PRIME did not change after retesting. Overall, Gram-positive bacilli, including *Corynebacterium*, exhibited higher accuracy in the ASTA MicroIDSys Elite (95.5%) compared to the VITEK MS PRIME (72.7–81.8%). The strains of *Corynebacterium jeikeium* and *Corynebacterium afermentans* that failed to be identified by the VITEK MS PRIME were both successfully identified at the species level by the ASTA MicroIDSys Elite.

### 3.3. Identification of Filamentous Fungi Using the IVD and RUO Databases

Among the 289 filamentous fungi, species-level identification using the IVD database was 71.3% (206/289), while genus- or complex-level identification was 75.4% (218/289) (Figure 1).

Additionally, when analyzed using the RUO database, species-level identification improved significantly to 85.8% (248/289), and genus- or complex-level identification increased to 88.2% (255/289). There was no significant difference in identification performance between the IVD and RUO databases for *Aspergillus* strains. However, for basidiomycetes, there was a significant improvement in identification performance when using the RUO database, with an increase from 35.3% to 100% (*p* < 0.0001). Detailed results for each strain are summarized in Table 3.

### 3.4. Identification Failure of Filamentous Fungi

A total of 34 isolates failed to be identified by the VITEK MS PRIME using both the IVD and RUO databases (Table 4); among them, 6 isolates were strains included in the IVD database, including *Aspergillus versicolor* (3), *Exophiala xenobiotica* (2), and *Nannizzia gypsea* (1). The remaining 28 isolates were strains included only in the RUO database, with *Penicillium* and *Talaromyces* strains accounting for 11 cases.

## 4. Discussion

Based on the species included in the VITEK MS IVD database, the performance of the VITEK MS PRIME for the identification of bacteria and yeasts using the PICKME was remarkable. It provided quick and reliable identifications for 89.4% of the isolates, using the same laboratory workflow as the conventional method in various bacterial and yeast species. A previous study highlighted the significant role of the PICKME in selecting rough *Nocardia* colonies and creating thin deposits, essential for accurate MALDI-TOF identification [9]. The study involved adding formic acid directly to the deposit along with the cyano-4-hydroxycinnamic acid (CHCA) matrix solution to facilitate bacterial wall destruction and protein extraction. Despite being an extra step, this direct deposit analysis method proved much faster than the sample preparation recommended by the manufacturer and was compatible with standard bacterial identification workflows.

The quality of the deposit on the target can influence the spectra’s quality. Proper training is needed to ensure the appropriate amount of colony transfer onto the plate [14]. Previous research reported significant differences in score distribution among testing institutions, even when using the same extraction method [15]. In this study, all unidentified isolates transferred with wooden toothpicks and 37.5% of unidentified isolates transferred with PICKME were correctly identified after retesting. Therefore, using the PICKME may lead to more consistent testing, unaffected by the examiner’s capabilities.

We did not observe a significant overall performance difference between the VITEK MS PRIME and ASTA MicroIDSys Elite. However, the VITEK MS PRIME exhibited relatively higher rates of inaccurate identification in the coryneform and aerobic/anaerobic Gram-positive bacilli groups, consistent with a previous study [16]. This difference may be attributed to the thick peptidoglycan layers, potentially interfering with laser ionization. Interestingly, among the 19 strains not detected by the VITEK MS PRIME, ASTA MicroIDSys Elite detected 7 strains at the species level and 3 strains at the genus level. Among the unidentified isolates in the VITEK MS PRIME, *Aeromonas dhakensis, Comamonas acidovorans, Sphingomonas paucimobilis*, and *Yarrowia galli* have been reported as unidentified isolates in previous systems and databases [13,17]. These species do not exist in VITEK MS IVD Database 3.2, but all of them have been reported as human pathogens [18,19,20,21]. Therefore, it is necessary to update the database in order to improve the ability to distinguish them from other species.

It is important to consider the impact of misidentification results in clinical laboratories. One of the advantages of the VITEK MS PRIME is its lower likelihood of causing confusion in result interpretation because it does not produce misidentification results. When identification is not possible, it reports as “no identification”, allowing the laboratory to promptly switch to alternative methods.

For filamentous fungi, the VITEK MS PRIME using the updated IVD database successfully gave species-level identification in 71.3% (206/289) of cases. The use of the RUO database significantly increased the identification ability to 85.8%, and especially basidiomycetes showed a remarkable improvement. The quality of the database is especially crucial for the identification of filamentous fungi using MALDI-TOF MS. Similar to our approach, previous studies reported improvements in identification accuracy achieved by integrating different databases [22,23]. This approach allowed for an enhanced identification process.

The manufacturers’ libraries for microbial identification are regularly updated to include more species and variations within individual species. However, they still face the challenge of incompleteness due to the vast diversity encountered in human pathology [14]. VITEK MS IVD Database 3.2 only covers 221 fungal species. While common molds are generally well covered, identifying rarer or cryptic species becomes more challenging. To improve the capability of identifying rare filamentous fungi, developing a homemade library or utilizing open public databases for analysis is possible, but it may not be straightforward to implement in actual laboratories [14]. The VITEK MS PRIME provides a practical solution, allowing a seamless transition to the RUO database for strains that fail in the existing IVD database without additional experiments. This is expected to significantly aid in the identification of rare filamentous fungi.

The VITEK MS PRIME successfully identified *T. rubrum* and *T. tonsurans* to the species level in IVD mode. However, although *T. interdigitale* and *T. mentagrophytes* form a species complex, differentiation between *T. interdigitale* and *T. mentagrophytes* was not achieved, even in RUO mode. This lack of accurate identification was also observed in a previous study using the VITEK MS v3.0 [24]. Indeed, some strains (*Aspergillus versicolor, Exophiala xenobiotica*, and *Nannizzia gypsea*) included in the IVD database could not be identified using the VITEK MS v3.2. These findings were also reported in a previous study, where Zvezdánova et al. reported correct identification rates of 86.4% for *Aspergillus* species and 65.7% for Mucorales [25]. It will be necessary to enhance the database for *Aspergillus*, Mucorales, and *Trichophyton* strains to improve their identification in the VITEK MS PRIME.

We believe our findings will be of interest to MALDI-TOF MS system users, especially in the clinical setting. We conducted the evaluation using both strains recommended for MALDI-TOF MS system evaluation in the guideline and strains obtained from real-world clinical status. For filamentous fungi, we have the privilege of an extensive archive of well-characterized fungi since filamentous fungus identification is particularly difficult in the clinical laboratory. We evaluated how well the new VITEK MS PRIME system identified our extensive list of fungi, utilizing not only the IVD database but also the RUO database. The identification rates in each database and the particularly difficult-to-identify strains can be referenced in the clinical laboratory when difficult filamentous fungus identification cases are encountered.

In this study, we showed that the VITEK MS PRIME with PICKME provided reliable results with high efficiency and allowed standardized sample preparation. For filamentous fungi, the VITEK MS PRIME provided reliable results for diverse filamentous fungi that are commonly isolated in clinical laboratories. For further identification, combining the use of the RUO database can be beneficial, especially for basidiomycetes.

## Figures and Tables

**Figure 1 microorganisms-12-00964-f001:**
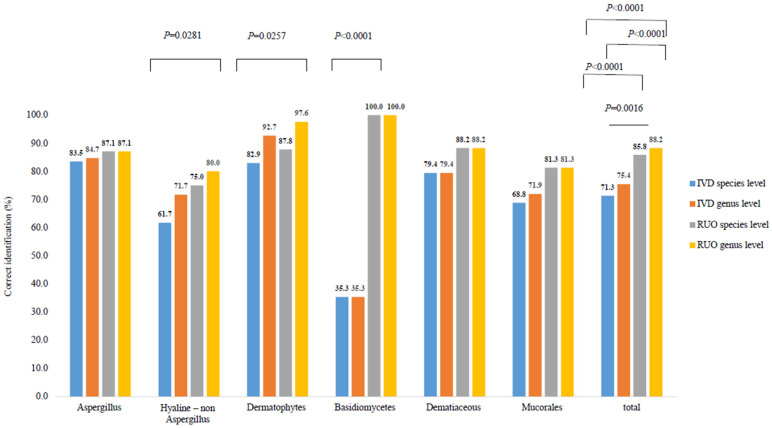
Correct identification percentages for molds by VITEK MS PRIME IVD database and RUO database (*n* = 289).

**Table 1 microorganisms-12-00964-t001:** Bacterial and yeast identification by VITEK MS PRIME and ASTA MicroIDSys Elite (*n* = 320).

	No. of Isolates	VITEK MS PRIME—Wooden Toothpick, *n* (%)			VITEK MS PRIME—PICKME, *n* (%)			ASTA MicroIDSys Elite, *n* (%)			Concordance Rate, *n* (%)		
		Correct ID (Species Level)	Incomplete ID (Genus Level)	No ID	Correct ID (Species Level)	Incomplete ID (Genus Level)	No ID	Correct ID (Species Level)	Incomplete ID (Genus Level)	No ID	Wooden Toothpick and PICKME	Wooden Toothpick and ASTA	PICKME and ASTA
Gram-positive cocci	91	90 (98.9)		1 (1.1)	89 (97.8)		1 (1.1)	87 (95.6)	1 (1.1)	3 (including 2 misID) * (3.3)	89 (97.8)	86 (94.5)	85 (93.4)
Gram-positive bacilli	22	18 (81.8)	1 (4.5)	3 (13.6)	16 (72.7)	1 (4.5)	5 (22.7)	21 (95.5)		1 (4.5)	16 (72.7)	18 (81.8)	16 (72.7)
Gram-negative cocci	15	15 (100)			15 (100)			13 (86.7)	1 (6.7)	1 (6.7)	15 (100)	13 (86.7)	13 (86.7)
Gram-negative bacilli	124	111 (89.5)	2 (1.6)	11 (8.9)	111 (89.5)	2 (1.6)	11 (8.9)	103 (83.1)	11 (8.9)	10 (8.1)	110 (88.7)	101 (81.5)	100 (80.6)
Anaerobes	16	14 (87.5)		2 (12.5)	14 (87.5)		2 (12.5)	14 (87.5)		2 (12.5)	14 (87.5)	14 (87.5)	14 (87.5)
Yeasts	52	46 (88.5)		6 (11.5)	44 (84.6)		8 (15.4)	44 (84.6)		8 (15.4)	42 (80.8)	41 (78.8)	39 (75.0)
Total	320	294 (91.9)	3 (0.9)	23 (7.2)	289 (90.3)		27 (8.4%)	282 (88.1)	13 (4.1)	25 (7.8)	286 (89.4)	273 (85.3)	267 (83.4)

* ASTA MicroIDSys Elite misidentified two *Staphylococcus cohnii* as *Micrococcus lylae.* Abbreviations: ID, identification; ASTA, ASTA MicroIDSys Elite; *n*, number.

**Table 2 microorganisms-12-00964-t002:** Discordant isolates between wooden toothpick and PICKME in VITEK MS PRIME (*n* = 34).

Discordant Isolates		VITEK MS PRIME—Wooden Toothpick		VITEK MS PRIME—PICKME		ASTA MicroIDSys Elite		Re-Tests
		Result	Score	Result	Score	Result	Score	
Identified Only in Wooden Toothpick (*n* = 8)								
*Aeromonas caviae*		*Aeromonas caviae*	99.9	No ID		*Aeromonas caviae*	233	
*Corynebacterium jeikeium*		*Corynebacterium jeikeium*	99.9	No ID		*Corynebacterium jeikeium*	178	
*Corynebacterium jeikeium*		*Corynebacterium jeikeium*	99.9	No ID		*Corynebacterium jeikeium*	178	
*Streptococcus* *anginosus*		*Streptococcus* *anginosus*	99.9	No ID		*Streptococcus* *anginosus*	211	
*Candida albicans*		*Candida albicans*	95.3	No ID		*Candida albicans*	139	
*Candida albicans*		*Candida albicans*	99.9	No ID		*Candida albicans*	224	Correct ID in PICKME
*Candida ciferrii*		*Candida ciferrii*	99.9	No ID		No ID		Correct ID in PICKME
*Candida parapsilosis*		*Candida parapsilosis*	99.9	No ID		*Candida parapsilosis*	141	Correct ID in PICKME
Identified only in PICKME (*n* = 3)								
*Acinetobacter* *nosocomialis*		No ID		*Acinetobacter nosocomialis*	90.7	No ID		Correct ID in wooden toothpick in VITEK
*Candida ciferrii*		No ID		*Candida ciferrii*	99.9	No ID		Correct ID in wooden toothpick in VITEK
*Candida tropicalis*		No ID		*Candida tropicalis*	99.9	*Candida tropicalis*	214	Correct ID in wooden toothpick in VITEK
Identified only at genus level (*n* = 4)								
*Acinetobacter* *proteolyticus *^#^*		No ID		*Acinetobacter gyllenbergii*	99.6	No ID		
*Aeromonas dhakensis *^#^*		*Aeromonas veronii/sobria*	50/50	*Aeromonas veronii/sobria*	50/50	No ID		
*Paenibacillus polymyxa*		*Paenibacillus peoriae*	99.9	*Paenibacillus peoriae*	99.9	No ID		
*Pseudomonas* *granadensis *^#^*		*Pseudomonas fluorescens*	99.7	No ID		No ID		
Misidentification (*n* = 19)								
*Acinetobacter seifertii* *^#^*		No ID		No ID		*Acinetobacter calcoaceticus*	137	
*Aerococcus viridans*		No ID		No ID		*Aerococcus viridans*	160	
*Aeromonas dhakensis* ****^#^*		No ID		No ID		*Aeromonas* sp.	195	
*Aeromonas dhakensis ** * ^#^ *		No ID		No ID		*Aeromonas* sp.	205	
*Bacteroides* *thetaiotaomicron*		No ID		No ID		No ID		Correct ID in ASTA
*Comamonas* *acidovorans* ***		No ID		No ID		*Comamonas* *acidovorans*	154	
*Corynebacterium* *afermentans*		No ID		No ID		*Corynebacterium* *afermentans*	169	
*Corynebacterium* *afermentans*		No ID		No ID		*Corynebacterium* *afermentans*	167	
*Eikenella corrodens*		No ID		No ID		Invalid ID		
*Fusobacterium* *nucleatum*		No ID		No ID		No ID		
*Gardnerella vaginalis*		No ID		No ID		Invalid ID		
*Pasteurella multocida*		No ID		No ID		*Pasteurella multocida*	141	
*Shewanella algae*		No ID		No ID		No ID		Correct ID in wooden toothpick & PICKME in VITEK
*Sphingomonas paucimobilis* **^#^*		No ID		No ID		Invalid ID		
*Stenotrophomonas maltophilia*		No ID		No ID		Invalid ID		Correct ID in ASTA
*Trichrosporon faecale *^#^*		No ID	No ID	No ID		Invalid ID		
*Yarrowia galli* **^#^*		No ID	No ID	No ID		Invalid ID		
*Candida dubliniensis*		No ID	No ID	No ID		*Candida dubliniensis*	153	Correct ID in wooden toothpick & PICKME in VITEK
*Candida tropicalis*		No ID	No ID	No ID		*Candida tropicalis*	153	

*** Species not included in VITEK MS IVD Database 3.2. ^#^ Species not included in ASTA MicroIDSys Elite CoreDB version 1.27.04. Abbreviations: ID, identification; No ID, No identification; Invalid ID, invalid identification; VITEK, VITEK MS PRIME; ASTA, ASTA MicroIDSys Elite.

**Table 3 microorganisms-12-00964-t003:** Filamentous fungus identification by VITEK MS PRIME IVD database and RUO database (*n* = 289).

		IVD DB			RUO DB		
Identification by DNA Sequencing	No. of Isolates	Species-Level ID	Genus- or Complex-Level ID	No ID	Species-Level ID	Genus- or Complex-Level ID	No ID
Hyaline—*Aspergillus* species							
*Aspergillus aculeatus **	2			2			2
*Aspergillus calidoustus*	2	2					
*Aspergillus chevalieri **	5			5			5
*Aspergillus flavus*	7	7					
*Aspergillus fumigatus*	7	7					
*Aspergillus japonicas **	1			1			1
*Aspergillus lentulus*	12	12					
*Aspergillus nidulans*	7	6		1	1		
*Aspergillus niger*	8	8					
*Aspergillus ochraceus*	1	1					
*Aspergillus sydowii*	14	13		1	1		
*Aspergillus tamarii*	2	1	1		1		
*Aspergillus terrei*	8	8					
*Aspergillus tubingensis*	6	6					
*Aspergillus versicolor*	3			3			3 *
Total	85	71	1	13	3		11
Hyaline—non-*Aspergillus* species							
*Acremonium sclerotigenum*	1	1					
*Beauveria bassiana*	1	1					
*Fusarium dimerum*	1	1					
*Fusarium equiseti **	2		1	1	1		1
*Fusarium oxysporum*	2	2					
*Fusarium solani*	4	4					
*Fusarium verticillioides*	4	1	3		2	1	
*Paecilomyces formosus **	4			4			4
*Paecilomyces variotii*	2	2					
*Penicillium chrysogenum*	3	3					
*Penicillium citrinum*	9	8		1	1		
*Penicillium crustosum **	1			1	1		
*Penicillium expansum **	1			1		1	
*Penicillium glabrum*	2	2					
*Penicillium janthinellum **	2			2		1	1
*Penicillium oxalicum **	4			4	3		1
*Penicillium toxicarium **	1			1			1
*Pseudallescheria boydii*	1	1					
*Purpureocillium lilacinum*	8	8					
*Rasamsonia argillacea*	3	3					
*Talaromyces marneffei **	1			1			1
*Talaromyces pinophilus **	2		2				2
*Talaromyces purpureogenus **	1			1			1
Total	60	37	6	17	8	3	12
Dermatophytes							
*Epidermophyton floccosum*	1	1					
*Microsporum canis*	10	9		1	1		
*Nannizzia gypsea*	1			1			1 *
*Trichoderma longibrachiatum*	2	2					
*Trichophyton erinacei*	1	1					
*Trichophyton interdigitale*	3	2	1			1	
*Trichophyton mentagrophytes*	3		3			3	
*Trichophyton rubrum*	17	16		1	1		
*Trichophyton tonsurans*	3	3					
Total	41	34	4	3	2	4	1
Basidiomycetes							
*Bjerkandera adusta*	1	1					
*Coprinellus radians **	4			4	4		
*Irpex lacteus*	11	11					
*Schizophyllum commune **	18			18	18		
Total	34	12		22	22		
Dematiaceous							
*Alternaria alternata*	11	11					
*Cladosporium cladosporioides*	6	6					
*Cladosporium halotolerans **	2			2	2		
*Colletotrichum gloeosporioides **	1			1			1
*Exophiala xenobiotica*	2			2			2 *
*Neoscytalidium dimidiatum **	1			1			1
*Scedosporium apiospermum*	10	10					
*Scopulariopsis brevicaulis **	1			1	1		
Total	34	27		7	3		4
Mucorales							
*Cunninghamella bertholletiae **	3			3			3
*Lichtheimia corymbifera*	3	3					
*Mucor circinelloides*	5	5					
*Mucor fragilis **	1		1				1
*Mucor irregularis **	1			1			1
*Mucor velutinosus*	1	1					
*Rhizomucor miehei **	1			1			1
*Rhizomucor pusillus **	4			4	4		
*Rhizopus microsporus*	3	3					
*Rhizopus oryzae*	10	10					
Total	32	22	1	9	4		6
Others							
*Eutypella scoparia*	3	3					
Total molds	289	206	12	71	42	7	34

* Species not included in VITEK MS IVD Database 3.2.

**Table 4 microorganisms-12-00964-t004:** Clinical filamentous fungus isolates that failed to be identified by VITEK PRIME MS (*n* = 34).

Isolates Included in IVD Database		No. of Isolates
*Aspergillus* species	*Aspergillus versicolor*	3
Dematiaceous	*Exophiala xenobiotica*	2
Dermatophytes	*Nannizzia gypsea*	1
**Isolates Included Only in RUO Database**		
*Aspergillus* species	*Aspergillus aculeatus*	2
	*Aspergillus chevalieri*	5
	*Aspergillus japonicus*	1
Hyaline—non-*Aspergillus*	*Fusarium equiseti*	1
	*Paecilomyces formosus*	4
	*Penicillium janthinellum*	1
	*Penicillium oxalicum*	1
	*Penicillium toxicarium*	1
	*Talaromyces marneffei*	1
	*Talaromyces pinophilus*	2
	*Talaromyces purpureogenus*	1
Dematiaceous	*Colletotrichum gloeosporioides*	1
	*Neoscytalidium dimidiatum*	1
Mucorales	*Cunninghamella bertholletiae*	3
	*Mucor fragilis*	1
	*Mucor irregularis*	1
	*Rhizomucor miehei*	1

## Data Availability

Data are contained within the article.

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
