# Peer review of "Clinical Evaluation of VITEK MS PRIME with PICKME Pen for Bacteria and Yeasts, and RUO Database for Filamentous Fungi"

_microorganisms, 2024, doi:10.3390/microorganisms12050964_

Round 1

Reviewer 1 Report

Comments and Suggestions for Authors

The article, titled "Clinical evaluation of VITEK MS PRIME with PICKME pen for Bacteria and Yeasts, and RUO database for Filamentous Fungi," submitted for review in the journal Microorganisms, presents a paper on the comparison of MALDI MS systems for identifying microorganisms. The study authors also used a new system for sample preparation. The manuscript is well written, however, I have a few reservations:

1. Information about bioMérieux's new system for identifying microorganisms appears in the Introduction. However, a general introduction to the MALDI method, systems from different manufacturers, and sample preparation methods is missing. Please complete the Introduction.

2. Among the strains tested, were there any species not included in one or both databases?

3. Please write an explanation (an additional paragraph) about the minor differences regarding the results that were obtained using systems in the Results/Discussion sections.

Author Response

  1. Information about bioMérieux's new system for identifying microorganisms appears in the Introduction. However, a general introduction to the MALDI method, systems from different manufacturers, and sample preparation methods is missing. Please complete the Introduction.

-> Thank you for your valuable comment. According to the reviewer`s comment, we added the information about the different MALDI TOF systems and sample preparation method in introduction (line 46-54).

“Currently, two manufacturers have received FDA approval and IVD certification for their MALDI-TOF MS systems: bioMérieux (Marcy-l’Étoile, France) and Bruker Daltonics (Bremen, Germany). And two new companies have joined the market, ASTA (Suwon, South Korea), and Zybio (Chongqing, China) [6]. Appropriate sample pretreatment meth-ods are crucial for the successful identification of microorganisms in MALDI-TOF system. Direct colony spotting and on-target extraction method are commonly used to identify bacteria, including gram-positive, gram-negative, and mucinous bacteria. In-tube extrac-tion method, a more complex and time-consuming approach, is used for microorganisms that are difficult to identify [7].”

  1. Among the strains tested, were there any species not included in one or both databases?

-> Thank you for your comment. According to the reviewer`s comment, we added the foot notes in species not included in VITEK MS IVD Database 3.2 and ASTA MicroIDsys Elite database in table 2 and table 3.

  1. Please write an explanation (an additional paragraph) about the minor differences regarding the results that were obtained using systems in the Results/Discussion sections

-> Thank you for your valuable comment. According to the reviewer`s comment, we modified the discussion as follows in line 227-243.

“We did not observe a significant overall performance difference between VITEK MS PRIME to ASTA MicroIDSys Elite. However, VITEK MS PRIME exhibited relatively higher rates of inaccurate identification in the coryneform and aerobic/anaerobic gram-positive bacilli group, consistent with a previous study [16]. This difference may be attributed to the thick peptidoglycan layer, potentially interfering with laser ionization. Interestingly, among the 19 strains not detected by VITEK MS PRIME, ASTA MicroIDSys Elite detected 7 strains at the species level and 3 strains at the genus level. Among the unidentified iso-lates in VITEK MS PRIME, Aeromonas dhakensis, Comamonas acidovorans, Spingomonas paucimobilis, and Yarrowia galli have been reported as unidentified isolates in previous systems and databases [13,17]. These species do not exist in the VITEK MS ver. 3.2 data-base, but all of them were reported as human pathogens [18-21]. Therefore, updating the database is needed to improve the distinguishing ability from other species.

It is important to consider the impact of misidentification results in clinical laborato-ries. One of the advantages of the VITEK MS PRIME is its lower likelihood of causing con-fusion in result interpretation due to not producing misidentification results. When identification is not possible, it reported as no identification allowing the laboratory to promptly switch to alternative methods.”

Reviewer 2 Report

Comments and Suggestions for Authors

Dear authors,

identification of fungi and yeasts is really a significant issue and requires scientific attention. However you have cited only 20 articles. Tables 1,2 and 3 are in a very particluar details which are not necessary for readers. There are small numbers of specific species in each table, are those statistically significant? Can we make correct conclusions? Are those results different from what is written in Biomerieux dossier? It is clear post-marketing study and I hardly can find scientific soundness.

Author Response

Thank you for your valuable comment. We believe our findings will be of interest to MALDI-TOF MS system users especially in the clinical setting for the below reasons, and we tried to address practical questions in the clinical laboratory in this study.

We have selected 228 isolates from 98 species from CLSI M58 lists and remaining 92 isolates were obtained from actual clinical samples requested to the laboratory. Therefore, we tried to represent both CLSI guideline and real-world clinical status. For bacteria and yeasts, we compared two MALDI-TOF MS systems, the new VITEK MS Prime system to our routinely used ASTA MicroIDSys system, to our knowledge the comparison data of these two systems is scarce. We evaluated the performance of VITEK-PICKME pen compared to our routine wooden toothpick because colony pick-and-smear quality and interpersonal variability are important analytical step in managing good quality results in MALDI-TOF MS testing. And the study does not try to validate the database, which is already done by the manufacturer in comparison to sequencing.

For filamentous fungi, we have the privilege of an extensive archive of well-characterized fungi since filamentous fungi identification is particularly difficult in the clinical laboratory. We evaluated how well the new VITEK MS Prime system identifies our extensive list of fungi, utilizing not only the IVD database but also the RUO database. The identification rates in each database and the particularly difficult to identify strains can be referenced by readers in the clinical laboratory when they encounter difficult filamentous fungi identification cases.

-> According to the reviewer`s comment, we have added the clinical impact of our study in discussion in line 274-282 as  

“We believe our findings will be of interest to MALDI-TOF MS system users especially in the clinical settings. We conducted the evaluation by strains recommended for MAL-DI-TOF MS system evaluation in the guideline and strains obtained from real-world clin-ical status. For filamentous fungi, we have the privilege of an extensive archive of well-characterized fungi since filamentous fungi identification is particularly difficult in the clinical laboratory. We evaluated how well the new VITEK MS Prime system identifies our extensive list of fungi, utilizing not only the IVD database but also the RUO database. The identification rates in each database and the particularly difficult to identify strains can be referenced in the clinical laboratory when they encounter difficult filamentous fungi identification cases.”

-> We clarify the bacteria and yeast species used in this study (line 80-89) as

“A total of 320 bacteria and yeast isolates were obtained from the International St. Mary`s hospital, consisting of 268 bacteria (124 gram-negative bacilli, 91 gram-positive cocci, 22 gram-positive bacilli, 16 anaerobes, 15 gram-negative cocci) and 52 yeasts (37 Candida species, 15 others). Among them, 228 isolates (76 isolates from 24 species for gram-negative bacilli, 60 isolates from 31 species for gram-positive cocci, 14 isolates from 8 species for anaerobes, 14 isolates from 8 species for gram-positive bacilli, 14 isolates from 7 species for gram-negative cocci, 50 isolates from 20 species for yeast) were selected from the list of strains recommended by Clinical and Laboratory Standards Institute (CLSI) guideline M58-ED1 [11]. The remaining 92 strains were obtained from clinical samples requested in the laboratory.”

-> According to the reviewer`s comment, we have more references in discussion as follows (line 247-251)  

“ The quality of the database is especially crucial for the identification of filamentous fungi using MALDI-TOF MS. Similar to our approach, previous studies were reported showed improvements in identification accuracy by integrating different databases [22,23]. This approach allowed for enhanced the identification process.”

  1. Jeraldine, V.M.; Wim, L.; Ellen, V.E. A comparative study for optimization of MALDI-TOF MS identification of filamentous fungi. Eur J Clin Microbiol Infect Dis 2023, 42, 1153-1161, doi:10.1007/s10096-023-04652-3.
  2. Trivitt, G.E.; Lau, A.F. Performance of the MSI-2 Database for Fungal Identification by Matrix-Assisted Laser Desorption Ionization-Time of Flight Mass Spectrometry from Cleanroom Environments. J Clin Microbiol 2023, 61, e0135322, doi:10.1128/jcm.01353-22.